# A Novel Multiepitope Fusion Antigen as a Vaccine Candidate for the Prevention of Enterotoxigenic *E. coli*-Induced Calf Diarrhea

**DOI:** 10.3390/vaccines12050457

**Published:** 2024-04-25

**Authors:** Haoyun Zhang, Xinwei Yuan, Yanfei He, Yingyu Chen, Changmin Hu, Jianguo Chen, Lei Zhang, Xi Chen, Aizhen Guo

**Affiliations:** 1National Key Laboratory of Agricultural Microbiology, Huazhong Agricultural University, Wuhan 430070, China; zhanghaoyun@webmail.hzau.edu.cn (H.Z.); yxw9766@webmail.hzau.edu.cn (X.Y.); heyanfei@webmail.hzau.edu.cn (Y.H.); chenyingyu@mail.hzau.edu.cn (Y.C.); hcm@mail.hzau.edu.cn (C.H.); chenjg@mail.hzau.edu.cn (J.C.); zhanglei2023@mail.hzau.edu.cn (L.Z.); 2Hubei Hongshan Laboratory, Wuhan 430070, China; 3College of Veterinary Medicine, Huazhong Agricultural University, Wuhan 430070, China; 4Key Laboratory of Development of Ruminant Bio-Products, Ministry of Agriculture and Rural Affairs, Wuhan 430070, China; 5The Cooperative Innovation Center for Sustainable Pig Production, Huazhong Agricultural University, Wuhan 430070, China

**Keywords:** ETEC, fimbriae, enterotoxin, multiepitope fusion antigen, vaccine

## Abstract

Calf diarrhea caused by enterotoxigenic *E. coli* (ETEC) poses an enormous economic challenge in the cattle industry. Fimbriae and enterotoxin are crucial virulence factors and vaccine targets of ETEC. Since these proteins have complicated components with large molecular masses, the development of vaccines by directly expressing these potential targets is cumbersome Therefore, this study aimed to develop a multiepitope fusion antigen designated as MEFA by integrating major epitopes of FanC and Fim41a subunits and a toxoid epitope of STa into the F17G framework. The 3D modeling predicted that the MEFA protein displayed the epitopes from these four antigens on its surface, demonstrating the desired structural characteristics. Then, the MEFA protein was subsequently expressed and purified for mouse immunization. Following that, our homemade ELISA showed that the mouse antiserum had a consistent increase in polyclonal antibody levels with the highest titer of 1:2^17^ to MEFA. Furthermore, the western blot assay demonstrated that this anti-MEFA serum could react with all four antigens. Further, this antiserum exhibited inhibition on ETEC adhesion to HCT-8 cells with inhibitory rates of 92.8%, 84.3%, and 87.9% against F17^+^, F5^+^, and F41^+^ ETEC strains, respectively. Additionally, the stimulatory effect of STa toxin on HCT-8 cells was decreased by approximately 75.3% by anti-MEFA serum. This study demonstrates that the MEFA protein would be an antigen candidate for novel subunit vaccines for preventing ETEC-induced diarrhea in cattle.

## 1. Introduction

Diarrhea is a significant disease in calves, with intricate causes, including pathogens and management factors, leading to substantial annual economic losses [1,2]. Enterotoxigenic *Escherichia coli* (ETEC) is one of the main bacterial pathogens responsible for newborn calf diarrhea [3,4]. Therefore, it is imperative to emphasize the prevention and control of ETEC-induced calf diarrhea in the cattle industry.

ETEC manifests two characteristic virulence factors: fimbriae and enterotoxins [5]. Fimbriae assist ETEC strains in attaching and colonizing the intestines, followed by the release of enterotoxin, which subsequently triggers diarrhea in calves [6]. Various types of fimbriae exist in ETEC, including F4, F5, F6, F17, F18, and F41 fimbriae [7]. For a long time, F5 and F41 fimbriae have been recognized as the predominant pathogenic factors in diarrheal ETEC, affecting different domestic animals [8,9]. In contrast, F17 fimbriae are associated with diarrhea and septicemia, commonly found in ETEC strains from ruminants [10]. Furthermore, recent epidemiological data have highlighted significant shifts in the prevalence of fimbrial types associated with calf diarrhea, shown by a decline in F5 and F41 fimbriae, while the emergence of F17 fimbriae was observed as the most common fimbriae in calf ETEC infection [11,12]. Although the association between F17 fimbriae and diarrhea is weaker than that between F5 or F41 fimbriae and diarrhea, changes in epidemic trends demonstrate the necessity to consider F17 fimbriae in vaccine strategies. The F5-F41 fimbriae vaccine against calf diarrhea has garnered early attention from researchers [13], and a multiple conjugate vaccine containing F5 has been studied in clinical practice and has demonstrated a favorable effect on calves [14]. However, the vaccines targeting F17 fimbriae are rarely reported. In addition, ETEC pathogenesis involves the production of enterotoxin [15]. The STa toxin often exhibits limited efficacy when immunized alone, but fusing it with other antigen proteins can enhance the immune response against STa [16]. A previous study has reported that a vaccine utilizing a recombinant adenoviral vector expressing the STa toxin with ETEC F5 fimbrial adhesin could effectively stimulate the production of antibodies targeting both the F5 fimbriae and STa [17]. Additionally, it has been observed that the fusion of STa toxoid epitopes with immunogenic protein subunits, such as the FanC subunit of F5 fimbriae or CfaB subunit of CFA adhesin, can also elicit the production of antibodies against the STa toxin [18,19]. However, no research has been reported on the potential of fusing F17 fimbriae with the STa enterotoxin as a potential vaccine candidate.

This study was aimed to develop a novel subunit vaccine against ETEC-induced diarrhea. Briefly, a genetic fusion strategy was utilized to integrate toxoid epitopes of STa and major B-cell epitopes from F5 and F41 fimbriae into the F17 fimbriae adhesion subunit F17G to develop a multiepitope fusion antigen designated as MEFA derived from the four proteins. This MEFA protein showed immunogenicity in mice and a reaction with anti-MEFA serum in vitro. Therefore, the MEFA protein exhibits the potential as a novel subunit vaccine against ETEC-induced diarrhea.

## 2. Materials and Methods

### 2.1. Animals

Specific-pathogen-free female BALB/c mice at the age of 6–8 weeks were purchased from and raised in the Center for Laboratory Animals in Huazhong Agricultural University (HZAU), China in strict accordance with the Guide for the Care and Use of Laboratory Animals, Hubei Province, China.

### 2.2. Bacterial Strains and Cells

The clinical strains F17^+^ ETEC HB-383, F41^+^ ETEC TC-44, and STa^+^ ETEC HB-502 were isolated, identified, and deposited at the National Key Laboratory of Agricultural Microbiology, HZAU. The standard F5^+^ ETEC strain C83529 and human adenocarcinoma (HCT-8) cells were obtained from the China Institute of Veterinary Drugs Control. These strains and cells were used for the in vitro evaluation of mouse antiserum.

### 2.3. Cloning of F17G, FanC, Fim41a, and STa Genes and Their Chimeric Gene F17G-Fim41a-FanC-STa

The coding sequence of F17G, FanC, Fim41a, and STa were amplified from strains F17+ ETEC HB-383, F5+ ETEC C83529, F41+ ETEC TC-44, and STa+ ETEC HB-502, respectively, using PCR. The PCR conditions were as follows: An initial denaturation step at 98 °C for 2 min was followed by 30 cycles of the following conditions: 10 s at 98 °C, 5 s at 56 °C, and 10 s at 72 °C. The amplification was completed with a final step of 72 °C for 1 min. The PCR products were verified by agarose gel electrophoresis and the bands were cut out. Then, these sequences were ligated separately to the pET30a vector digested by *Nde* I and *Sal* I enzymes using homologous recombination. The PCR primers used to amplify the fimbriae subunit genes are listed in Table 1. Meanwhile, the chimeric gene F17G-Fim41a-FanC-STa was designed as follows: The sequence from the F17G subunit of F17 fimbriae refers to the NCBI F17G coding sequence (AF055313.1). Epitopes of F17G were identified using a protein analysis software DNAstar and a network-based epitope prediction program [20,21]. These epitopes were categorized into major and minor epitopes based on their scores and frequencies. Highly antigenic major epitopes were retained to induce anti-F17G immunity. Three minor epitopes were replaced by major epitopes of the Fim41a (F41 fimbriae) subunit, FanC (F5 fimbriae) subunit, and toxoid epitope of STa, respectively. Various combinations were explored during the design process, and multiple combined protein sequences were input into the protein 3D modeling program Rosetta and PyMOL to predict the conformation of the new chimeric protein. A model with the desired structure exposing each epitope was selected for further research. Finally, a 1062 bp nucleotide sequence based on the model was synthesized into pET30a by Tsingke Biotec (Wuhan, China) The recombinant expression plasmid pET30a-F17G-Fim41a-FanC-STa was abbreviated as pET30a-MEFA.

### 2.4. Expression of the Recombinant Proteins

The recombinant plasmids pET30a-MEFA, rF17G, rFanC, rFim41a, and rSTa were transformed into the *E. coli* strain BL21. To express MEFA, rF17G, rFanC, rFim41a, and rSTa proteins, these strains were inoculated into LB containing kanamycin (50 μg/mL) and cultured at 37 °C to OD_600nm_ = 0.6. The cultures were then induced with IPTG (1 mM) at 37 °C for 6 h. Bacteria were collected by centrifugation at 10,000× *g* for 15 min, and the cells were broken at 800 bar for 5 cycles using a high-pressure homogenizer (Life Technologies, Carlsbad, CA, USA) after being suspended in PBS. The fraction of the inclusion body was purified by Ni column affinity chromatography using PBS containing 8 M urea. The purified His-tagged proteins were dialyzed in PBS with gradually decreasing urea concentrations (6 M, 4 M, 2 M, and 0 M). Finally, the protein was dialyzed overnight in urea-free PBS and collected. The purified MEFA, rF17G, rFanC, rFim41a, and rSTa proteins were analyzed with 12% or 15% SDS-PAGE gel. After being stained with coomassie blue and destained, the gels were photographed with an imaging system (Bio-Rad, Hercules, CA, USA). These proteins were kept for the development of mouse antiserum and reaction detection with iELISA and western blotting assay (WB).

### 2.5. Development of Mouse Antiserum

Ten female BALB/c mice aged 6 to 8 weeks were randomly divided into two groups, each containing five mice. The immunized group received a subcutaneous injection of 100 μg of MEFA protein in 200 μL mixed with Freund’s complete adjuvant (Sigma, St. Louis, MO, USA). This was followed by two boosters with the same dose but mixed with Freund’s incomplete adjuvant (Sigma, St. Louis, MO, USA) every two weeks. The control group followed the same injection procedure using PBS. Blood samples were collected from each mouse at 0, 14, 28, and 42 days. After 2 h standing at RT, the blood was centrifuged at 4000× *g* for 5 min to collect serum and then stored at −20 °C.

### 2.6. Antibody Detection with iELISA and WB Assay

The conventional home-established indirect enzyme-linked immunosorbent assay (iELISA) was used to detect anti-MEFA, anti-rF5, anti-rF17, anti-rF41, and anti-rSTa antibodies in the mouse sera. Briefly, the proteins MEFA, rF17G, rFanC, rFim41a, and rSTa were used as ELISA-coating antigens to determine the antibody levels in the serum. Each protein was diluted with a desired concentration of antigen-coating buffer (0.015 M NaCO_3_ and 0.035 M NaHCO_3_, with pH 9.6). Proteins were coated in 96-well plates with 100 μL (1 ng/μL) per well and incubated at 4 °C overnight. The plates were washed three times with PBS containing 0.05% Tween-20 (PBST) and blocked at 37 °C for 1 h with 200 μL of PBST containing 10% skim milk. After washes, each well was incubated at 37 °C for 1 h with serum samples from immunized or control mice (diluted with PBST to different concentrations containing 2.5% skim milk power). The wells were washed three times with PBST, and 100 μL of horseradish peroxidase (HRP)-conjugated goat anti-mouse IgG (1:10,000) was added, followed by incubation at 37 °C for 1 h. The wells were washed three times with PBST, and the color was developed with 100 μL of TMB Single-Component Substrate solution (Solaibio Scientific, Beijing, China) at RT for 10 min. Finally, OD_450nm_ was measured with FLUOSTAR OMEGA (BMG LABTECH, Offenburg, Germany) after the termination of the reaction. Additionally, antibodies were classified using a mouse antibody typing kit (Biodragon Biotech, Beijing, China).

Meanwhile, WB was used to detect specific responses between the anti-MEFA serum and the MEFA, rFanC, rF17G, rFim41a, and rSTa proteins. Briefly, each protein (10 μg) was separated by 12% or 15% sodium dodecyl sulfate polyacrylamide gel electrophoresis (SDS-PAGE) and transferred onto a PVDF membrane. The PVDF membrane was blocked overnight at 4 °C with 5% milk-TBST and washed with TBST three times, 15 min each time. Either anti-His tag mouse monoclonal antibody (1:5000) (Abbkine Scientific, Wuhan, China) or homemade anti-MEFA serum (1:2000) as the primary antibodies were added, incubated at room temperature (RT) for 2 h, and washed again with TBST. Horseradish peroxidase (HRP)-conjugated goat anti-mouse IgG (1:10,000) (Abbkine Scientific, Wuhan, China) was added and incubated at RT for 2 h. ECL luminescence was used to visualize the recombinant proteins by ChemiDoc Touch (Life Technologies, Carlsbad, CA, USA).

### 2.7. Adherence and Adherence Inhibition Assays

Adhesion and adhesion inhibition experiments were performed using F17^+^, F5^+^, and F41^+^ ETEC strains. Following the method described previously [18], HCT-8 cells (1 × 10^5^/well) were inoculated into 12-well tissue culture plates containing Dulbecco Modified Eagle Medium (DMEM) supplemented with 10% fetal bovine serum. F17^+^, F5^+^, and F41^+^ ETEC strains were cultured overnight at 37 °C, and then the bacterial solution was counted using a plate and added to each well after treatment. Specifically, a bacterial suspension (100 μL) (5 × 10^5^ CFU) was added to each well (1 × 10^5^ cells/well) at the multiplicity of infection (MOI) = 5. For the inhibition assay, the same ETEC suspension (100 μL) was incubated with anti-MEFA serum (20 μL) or pre-immune serum at RT for 1 h. Then the mixtures were added into wells and placed in an incubator at 37 °C (5% CO_2_) for 1 h. Each well was subsequently washed three times with PBS to remove non-adherent bacteria. The cells were then incubated with 0.25% trypsin (200 μL) in an incubator at 37 °C (5% CO_2_) for 30 min. The digested cells were collected by centrifugation at 12,000× *g* for 5 min, and the resulting pellets were resuspended in PBS. After dilution, the suspension was cultured on LB agar at 37 °C overnight to enumerate ETEC strains. The inhibition rate for adherence was calculated as follows: inhibition rate = (adhered bacteria number after treatment with pre-immune serum − adhered bacteria number after treatment with anti-MEFA serum)/(adhered bacteria number after treatment with pre-immune serum) × 100%.

### 2.8. The Activating Impact and Neutralizing Properties of the Antibody

The stimulatory effect of the STa toxin on cells was evaluated by measuring the cyclic guanosine monophosphate (cGMP) level in HCT-8 cells as follows [22]. HCT-8 cells were also introduced into 12-well culture plates as described above. For the neutralization activity assay, the STa toxin (2 ng) was incubated with anti-MEFA serum (100 μL) at RT for 1 h, and then, mixtures were added to each well. For the stimulatory effect assay, pre-immune serum was used as a positive control instead of anti-MEFA serum. Furthermore, the wells containing an equivalent volume of PBS were utilized as a blank control. After being cultured at 37 °C for 2 h, the cells were washed three times with PBS and lysed with 0.25% trypsin, as previously described. The concentration of cGMP in the lysed supernatant was measured according to the manufacturer’s instructions using the Human Cyclic Guanosine Monophosphate (cGMP) ELISA Kit (MLbio Technology, Shanghai, China). Furthermore, the stimulatory effect of the STa^+^ ETEC strain on HCT-8 cells was also evaluated as described above. All tests were conducted in triplicate simultaneously. The inhibition rate for stimulation was calculated as follows: inhibition rate = (cGMP level after treatment with pre-immune serum − cGMP level after treatment with anti-MEFA serum)/(cGMP level after treatment with pre-immune serum) × 100%.

### 2.9. Statistical Analysis

All statistical analyses were carried out using SPSS software (Version 21.0, SPSS, Chicago, IL, USA), and the results were presented as means ± standard errors. A two-tailed Student’s *t* test was employed to compare the experimental and control groups. Statistical significance was determined to be significant with *: *p* < 0.05, **: *p* < 0.01, ***: *p* < 0.001.

## 3. Results

### 3.1. Computational Modeling of the MEFA Protein

The F17G subunit was employed as the structural framework, and the epitopes of the Fim41a, FanC, and STa subunits were introduced at different insertion sites. Then, 3D models were generated for the simulation of the MEFA protein using the Rosetta 3.13 and PyMOL 2.6.0 programs and compared with the backbone protein F17G. The structure of the model with the top conformer score was used for the following research. Specifically, the major epitope of Fim41a (EKLMPGQSASTSYSGFHNWDDLSHRNYTSANKA) replaced F17G amino acid residues 90–99, F17G amino acid residues 201–213 were replaced by the major epitope of FanC (TNVGNGSGGANINTSFTTA), and the STa toxoid epitope (NNTFYCCELCCSPACAGC) [22] replaced F17G amino acid residues 227–245 (Figure 1A). The modeling results demonstrated that the epitopes of the FanC, Fim41a, and STa subunits were all exposed on the surface of the MEFA protein (Figure 1B).

### 3.2. Expression of the Recombinant Proteins

The SDS-PAGE gel showed the presence of the His-tagged MEFA, rF17G, rFanC, rFim41a, and rSTa proteins with expected sizes (Figure 2A). WB analysis demonstrated that the purified His-tagged MEFA, rF17G, rFanC, rFim41a, and rSTa proteins were specifically bound to the anti-His tag mouse monoclonal antibody (Figure 2B). This confirmed the correct expression of proteins.

### 3.3. Antibody Detection with iELISA and WB Assay

The results of WB and iELISA indicated that immunized mice generated an immune response to the MEFA protein, with significantly increased antibody titers after two booster vaccinations (Figure 3A). In contrast, no antibody response to these antigens was detected in the serum before the primary immunization and in the serum of control mice (Figure 3A). The anti-MEFA serum exhibited reactivity with the recombinant MEFA, rF17G, rFim41a, rFanC, and rSTa proteins in WB assays (Figure 3B). Furthermore, the MEFA protein induced the potent and specific anti-MEFA, anti-F17G, anti-Fim41a, anti-FanC, and anti-STa IgG antibodies in the antiserum. After the second booster, specific antibodies against these antigens could be detected even if the serum was diluted over 10,000-fold (Figure 3C). The results of antibody subtyping showed that the major antibody subtype was IgG1, followed by IgG2b after immunization (Figure 3D).

### 3.4. Adherence and Adherence Inhibition Assay

The adhesion of F17^+^, F5^+^, and F41^+^ ETEC strains to HCT-8 cells was significantly inhibited after incubation with anti-MEFA serum (Figure 4). The inhibition rates against F17^+^, F5^+^, and F41^+^ ETEC strains were 92.8%, 84.3%, and 87.9%, respectively.

### 3.5. The Activating Impact and Neutralizing Properties of the Antibody

The level of cGMP was employed as an indicator of STa toxin stimulation in HCT-8 cells. The presence of anti-MEFA serum was found to neutralize the STa toxin and further diminish the level of cGMP in HCT-8 cells. Specifically, the cGMP concentration in HCT-8 cells was 2.496 ± 0.106 pmol/mL when cells were co-incubated with the STa toxin and anti-MEFA serum. This value was significantly lower compared to the cGMP concentration (10.001 ± 0.716 pmol/mL) in cells co-incubated with STa toxin and pre-immune serum (Figure 5A). The stimulatory effect of HCT-8 cells by the STa toxin was observed to decrease by approximately 75.3%. Additionally, it was observed that anti-MEFA serum reduced the stimulatory impact of the STa^+^ ETEC strain on HCT-8 cells (Figure 5B). The anti-MEFA serum prevented the STa^+^ ETEC strain from having a stimulatory effect on HCT-8 cells, with a decrease of approximately 32.0%.

## 4. Discussion

ETEC represents a prominent etiological agent responsible for calf diarrhea, typically occurring within the first month of life, resulting in either immediate mortality or a grim prognosis for the infected calves [23]. The presence of antigenic diversity presents a significant challenge in preventing ETEC infections due to the multitude of virulence factors and the absence of cross-protective immunity [24]. Numerous efforts have been undertaken to develop vaccines against ETEC in pigs and humans [25,26], but there is a dearth of research on bovine ETEC vaccines, with most studies focusing on F5 fimbriae [27]. Nevertheless, a meta-analysis has revealed a progressive decline in the prevalence of F5 and F41 fimbriae in cases of calf diarrhea, while F17 fimbriae has emerged as the predominant type [11]. Although the association between F17 and clinical disease is weaker than for F5 and F41, this changing landscape underscores the necessity of reevaluating vaccine strategies, emphasizing newer and more prevalent fimbrial antigens to ensure the effectiveness of vaccines for preventing ETEC in calves. In addition, F17 fimbriae are also expressed by a few human uropathogenic *E. coli* strains, where they are known as G fimbriae, suggesting a certain public health risk [28]. Therefore, it reminds us to consider F17 fimbriae as an alternative target for ETEC vaccines. F17 fimbriae consist of the structural subunit F17A and adhesion subunit F17G [29]. Both the structural and adhesion subunits of other fimbriae, which belong to the same type, have been reported to exhibit equivalent immunogenic efficacy [30]. In unpublished data, we confirmed that mice immunized with F17G protein could resist the F17^+^ ETEC challenge. Compared to F17A, F17G may offer broader protective efficacy with fewer subtypes and higher homology [28]. Thus, the adhesion subunit F17G was used as the foundation to synthesize the MEFA protein in this study.

Additionally, enterotoxin is another essential virulence factor of ETEC. The STa toxin could activate the production of cyclic adenosine monophosphate (cAMP) and cGMP of small intestinal epithelial cells, resulting in fluid and electrolyte imbalances and the subsequent manifestation of diarrhea [31]. Another study has indicated that augmenting the quantity of STa toxoid epitopes in the fusion protein can notably enhance the levels of anti-STa antibodies [22]. However, given that the number of epitopes for substitution in the skeleton protein is limited and excessive modifications to the structure of F17G may have a potential impact, only one copy of the STa toxoid epitope was opted to be inserted in this study.

Data from our WB and ELISA assays indicate that the anti-MEFA serum demonstrated specific reactivity not only toward MEFA but also to its four parental proteins rF17G, rFanC, rFim41a, and rSTa, indicating that the epitopes in the protein were adequately exposed and elicited the corresponding antibodies. A similar result was shown by another multiepitope fusion protein from porcine ETEC, but our study elicited a higher level of anti-FanC and anti-Fim41a antibodies compared to the previous study [32]. The antibody response against F17G was observed to be the most pronounced. This may be partially due to the fact that the F17G backbone retains more epitopes in the MEFA protein. In contrast, the antibodies targeting STa exhibited substantially lower levels compared to other antigens. This discrepancy may be attributed to variations in epitope length and extent of exposure.

To further substantiate the neutralizing activity of anti-MEFA serum, we conducted an adherence inhibition assay and a stimulation assay by utilizing HCT-8 cells in vitro. The HCT-8 cell line, a human colorectal cancer cell line, was used because we were not able to obtain a standard bovine intestine cell line. In unpublished data, we evaluated the sensitivity of ETEC adhesion and STa toxin stimulation to HCT-8 cells, Madin–Darby bovine kidney (MDBK) cells, and embryonic bovine lung (EBL) cells and found that HCT-8 cells are more sensitive to ETEC adhesion and STa toxin stimulation compared to other cells. Therefore, the HCT-8 cell line is a more suitable alternative to other cell lines for in vitro studies. Adhesion experiments demonstrated that the adherence of F17^+^, F5^+^, and F41^+^ ETEC strains to HCT-8 cells could be significantly inhibited by anti-MEFA serum, with the most pronounced inhibitory effect on F17^+^ ETEC strain. This result may be attributed to the fact that the adhesin subunit F17G possesses the most intact epitope as a skeleton protein. Similarly, the adhesin subunit FedF of F18 fimbriae has been shown to induce neutralizing antibodies in mice against the corresponding ETEC [32]. Consequently, it is a viable approach to employ adhesin subunits as vaccine targets specifically against ETEC with associated fimbriae.

Further, we assessed the impact of STa toxin on HCT-8 cells. The STa toxin induces the production of cAMP and cGMP in host intestinal epithelial cells, resulting in diarrhea [33]. Therefore, the level of cGMP can serve as a direct indicator of the extent of STa stimulation on cells. Notably, the stimulation of HCT-8 cells was significantly decreased when STa toxin was neutralized by anti-MEFA serum, which is in accordance with a previous study on STa demonstrating that toxoid epitopes elicited modest antibody level and substantial neutralizing activity against STa toxin [18]. Moreover, the inhibitory effect of anti-MEFA serum on the stimulation of the STa^+^ ETEC strain (Figure 5) was significantly weaker than that of STa toxin stimulation. This fact may be attributed to the persistent production of the STa toxin by a viable ETEC strain, which hinders the complete neutralization by anti-MEFA serum.

Unfortunately, the current study did not extend the findings of the MEFA protein against ETEC to in vivo study. In view of practical applications, only the MEFA protein is not sufficient to protect calves from diarrhea. As well known, calf diarrhea is characterized by a multifactor involvement including several pathogens in the form of co-infections or secondary infections [34,35]. Therefore, the ideal subunit vaccines should incorporate the multiple antigenic proteins or epitopes from multiple pathogens associated with calf diarrhea.

## 5. Conclusions

In summary, this study developed an artificial MEFA protein derived from F5^+^, F17^+^, F41^+^, and STa^+^ ETEC. The anti-MEFA polyclonal antibodies could react with all four parental proteins rF17G, rFim41a, rFanC, and rSTa and effectively impeded F5^+^, F17^+^, and F41^+^ ETEC adhesion and alleviated the stimulatory impact of the STa toxin on cells in vitro. These results suggest the potential of the MEFA protein as a vaccine candidate against ETEC-induced calf diarrhea. Furthermore, the utilization of MEFA-based structural vaccinology may present a promising approach for the creation of a multivalent vaccine targeting other pathogens responsible for calf diarrhea.

## Figures and Tables

**Figure 1 vaccines-12-00457-f001:**
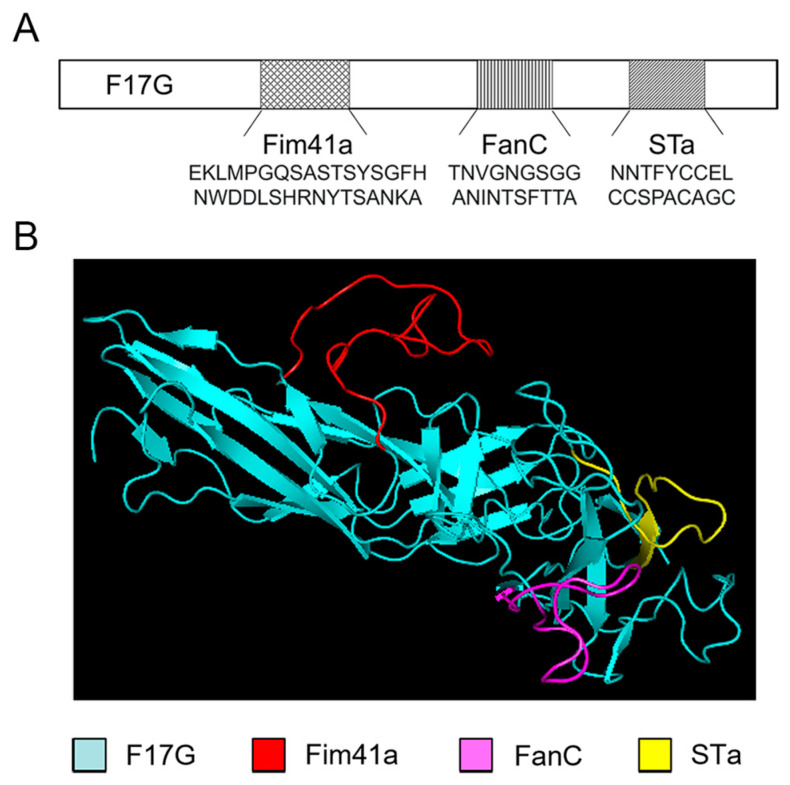
Design and computer modeling of the multiepitope fusion antigen (MEFA) protein. (**A**) Schematic diagram of the MEFA protein structure. Three low antigenicity epitopes of F17G were replaced by epitopes of Fim41a, FanC, and STa subunits. The amino acid sequences of the epitopes are shown in corresponding positions in the diagram. (**B**) The F17G-Fim41a-FanC-STa MEFA model was constructed using Rosetta.

**Figure 2 vaccines-12-00457-f002:**
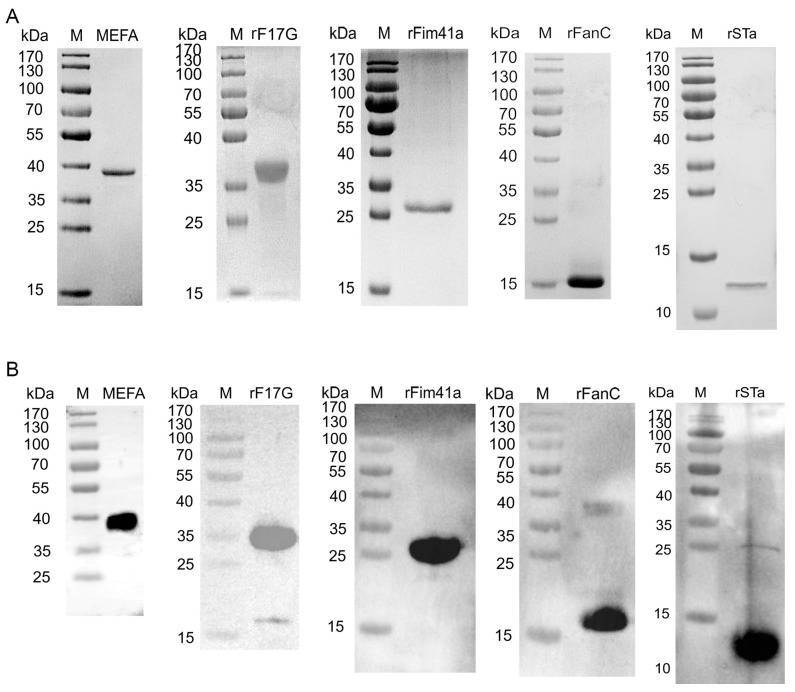
Analysis of purified His-tagged MEFA, rF17G, rFim41a, rFanC, and rSTa proteins using 12% or 15% SDS-PAGE and WB. (**A**) Detection of purified His-tagged MEFA, rF17G, rFim41a, rFanC, and rSTa proteins by 12% or 15% SDS-PAGE. (**B**) The purified His-tagged MEFA, rF17G, rFim41a, rFanC, and rSTa proteins were transferred onto a PVDF membrane and incubated with an anti-His tag mouse monoclonal antibody (1:5000). M: Prestained protein markers.

**Figure 3 vaccines-12-00457-f003:**
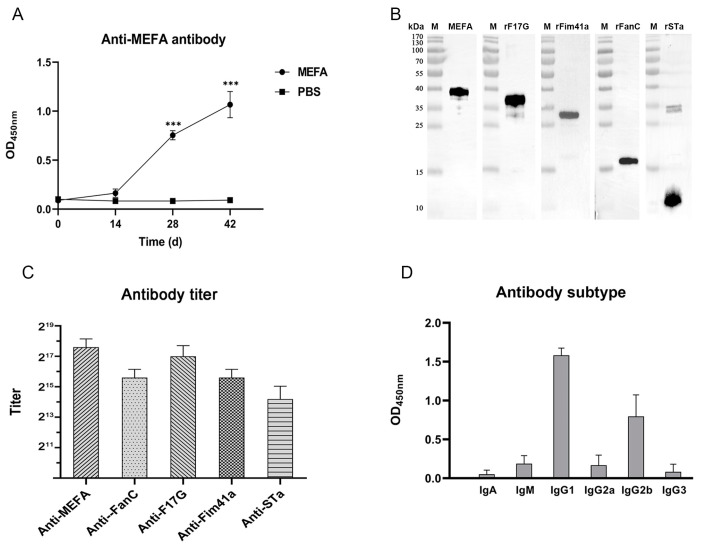
Antibody detection of mice immunized with MEFA, rF17G, rFim41a, rFanC, and rSTa proteins. (**A**) Antibody level in the anti-MEFA serum (***: *p* < 0.001, compared to the PBS group). (**B**) The reaction between the anti-MEFA serum with MEFA, rF17G, rFim41a, rFanC, and rSTa proteins detected by western blotting assay. (**C**) The antibody titer of the anti-MEFA serum was analyzed using the homemade iELISA (the criterion was P/N ≥ 2.1). (**D**) Antibody subtypes of anti-MEFA serum.

**Figure 4 vaccines-12-00457-f004:**
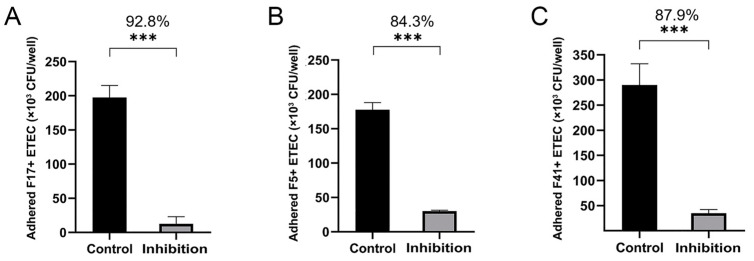
The inhibitory effect of the anti-MEFA antibody on the adherence of F17^+^, F5^+^, and F41^+^ ETEC strains to HCT-8 cells. (**A**–**C**) The adhesion of F17^+^, F5^+^, and F41^+^ ETEC strains to HCT-8 cells was significantly inhibited after pretreatment with anti-MEFA serum (***: *p* < 0.001).

**Figure 5 vaccines-12-00457-f005:**
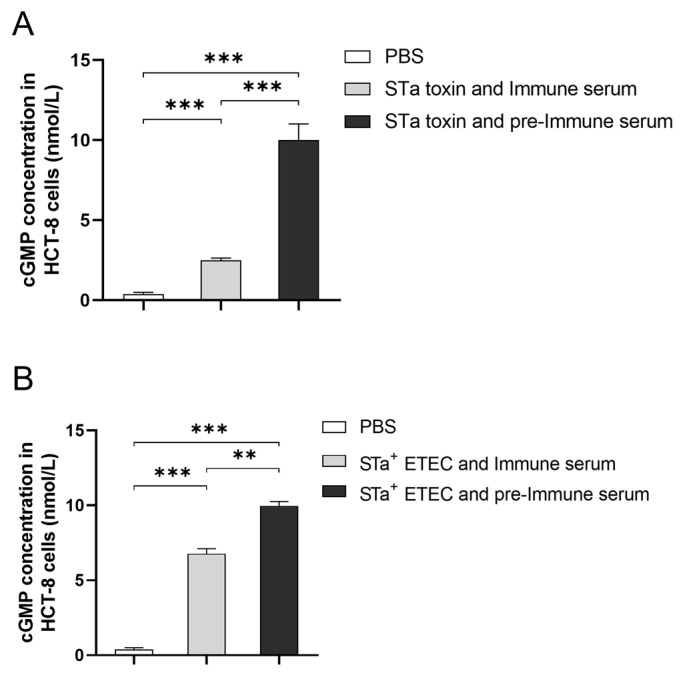
Antibody neutralization of the STa toxin. (**A**) Anti-MEFA serum significantly reduced STa toxin stimulation of intracellular cGMP in HCT-8 cells. (**B**) Anti-MEFA serum partially inhibited STa+ ETEC strain stimulation of intracellular cGMP in HCT-8 cells (**: *p* < 0.01, ***: *p* < 0.001).

**Table 1 vaccines-12-00457-t001:** Primers used in this study.

Primers	Sequences (5′–3′)
F17G-F	CTTTAAGAAGGAGATATACATATGGCGGCAGTTTCATTTATTGGTGC
F17G-R	GGCCGCAAGCTTGTCGACCTGATAGGAAAATGTAAATG
FanC-F	CTTTAAGAAGGAGATATACATATGAATACAGGTACTATTAAC
FanC-R	TGCGGCCGCAAGCTTGTCGACCATATAAGTGACTAAGAAGG
Fim41a-F	CTTTAAGAAGGAGATATACATATGGCTGCTGATTGGACGGAAGG
Fim41a-R	TGCGGCCGCAAGCTTGTCGACACTATAAATAACGGTGATAG
STa-F	CTTTAAGAAGGAGATATACATATGATGAAAAAGCTAATGTTGGC
STa-R	TGCGGCCGCAAGCTTGTCGACATAACATCCAGCACAGGC

## Data Availability

The datasets generated for this study are available upon request to the corresponding author.

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
