# Peer review of "A Novel Multiepitope Fusion Antigen as a Vaccine Candidate for the Prevention of Enterotoxigenic E. coli-Induced Calf Diarrhea"

_vaccines, 2024, doi:10.3390/vaccines12050457_

Round 1
Reviewer 1 Report
Comments and Suggestions for Authors
The paper reports the design and production, via recombinant microbial synthesis, of a novel protein (MEFA) bearing epitopes of three common ETEC fimbria (F5[K99], F17 and F41), plus the heat-stable enterotoxintoxin STa. As a candidate immunogen for a subunit vaccine against ETEC disease in calves, MEFA was tested for its ability to stimulate antibody responses in mice to each of its component elements. Assays in vitro using human adenocarcinoma (HCT-8) cell cultures were used to assess the activity of the murine antisera so generated in suppressing adhesion of ETEC bearing F5, F17 or F41, and in suppressing the effects of STa, as measured by intracellular cGMP.
The paper is concise, generally well-organised and clear. It reports an innovative approach to generating multivalent subunit immunogenic molecules, potentially for use in vaccines.
There are a number of minor issues and errors, listed below, that need attention.
The first part of the introduction (up to line 45 and references 1-9) uses very recent papers reporting small scale or very specific findings, to reference a general introduction to calf diarrhoea, including ETEC. There is foundational and review literature which, although less recent, is more suitable as reference material for this section.
Line 49-50: Although F17 fimbriae are more commonly reported in the cited references, there is a far weaker association between F17 and diarrhoea, than there is between F5 or F41 and diarrhoea. I think this should be acknowledged.
Line 67: You use the term 'F17G' here, but it is not stated that this is the adhesion subunit of F17 until the Discussion - please make this fact clear at this introductory stage
Line 113-4: You need to provide some detail on the conditions and apparatus used to lyse the bacteria under pressure. The phrase 'The fraction of inclusion body ...' is unclear. Please briefly describe how the cell debris was treated before affinity purification.
Line 115: Suggest you describe the proteins as His-tagged here - it makes it easier for non-specialist readers to understand the affinity purification and the subsequent use of anti-his anibody for labelling.
Line 117-8: "The purified proteins MEFA, rF17G, rFanC, rFim41a, and rSTa were purified ..." Is this just inappropriate repetition of 'purified' in this sentence, or was there an additional purification step?
Line 126: Why did you not use adjuvant without MEFA as your control injection, instead of PBS?
Line 127: "obtained and" is inappropriate here.
Lines 113-4: I think 'coating' is more appropriate than 'coated' in both instances
Line 184: Was the STa you used the his-tagged recombinant protein you had purified, or did you obtain it elsewhere?
Line 188: were the cells lysed with trypsin, as previously described?
Lines 223-5: This procedure is not described in Materials and Methods - you should transfer most of this description into M&M.
Line 249: Fig 3 title should be expanded to include immunisation with FanC, Fim41a,F17G & STa, as well as MEFA.
Line 250: suggest you retain only P<0.0001, as this is the only significance level marked
Line 258: Consider listing inhibition % to 2 sig figs (i.e. 93%, 84%, 88%) - it seems more appropriate and is more readable.
Line 263: as per comment for line 250
Figure 5: The diagonal hatching on the PBS control columns is difficult to distinguish from the hatching on the STa+preimmune serum columns. Suggest the PBS columns are a solid shade (grey, white or black).
Line 277: Change "Antibodies neutralizated STa toxin " to 'Antibody neutralization of STa toxin".
Lines 289-91: Again, you should acknowledge that the association between F17 and clinical disease is weaker than for F5 and F41.
Line 321: Is this a technical (statistical) use of the word 'significantly' - if so, I can't see evidence of this in the results. If not, I advise using a different word, for clarity, e.g. 'substantially'
Line 331: "The HCT-8 line is a suitable alternative" ... to what? To in vivo studies? To other cell lines? The sentence needs re-phrasing for clarity.
Lines 357-361: This is accidentally-incorporated extraneous text, I think.
Comments on the Quality of English LanguageThe English is of a very high standard. There are one or two inappropriate word variants used (commented on), and the very occasional typo or mis-spelling that should be resolved in proof-setting and proof-reading.
Author Response
Thank you for your comments. Below you will find our point-by-point responses In the attachment.
The paper reports the design and production, via recombinant microbial synthesis, of a novel protein (MEFA) bearing epitopes of three common ETEC fimbria (F5[K99], F17 and F41), plus the heat-stable enterotoxintoxin STa. As a candidate immunogen for a subunit vaccine against ETEC disease in calves, MEFA was tested for its ability to stimulate antibody responses in mice to each of its component elements. Assays in vitro using human adenocarcinoma (HCT-8) cell cultures were used to assess the activity of the murine antisera so generated in suppressing adhesion of ETEC bearing F5, F17 or F41, and in suppressing the effects of STa, as measured by intracellular cGMP.
The paper is concise, generally well-organised and clear. It reports an innovative approach to generating multivalent subunit immunogenic molecules, potentially for use in vaccines.
There are a number of minor issues and errors, listed below, that need attention.
The first part of the introduction (up to line 45 and references 1-9) uses very recent papers reporting small scale or very specific findings, to reference a general introduction to calf diarrhoea, including ETEC. There is foundational and review literature which, although less recent, is more suitable as reference material for this section.
A: Thank you for the comment. References [1-8] have been revised. Please see lines 390-404.
Line 49-50: Although F17 fimbriae are more commonly reported in the cited references, there is a far weaker association between F17 and diarrhoea, than there is between F5 or F41 and diarrhoea. I think this should be acknowledged.
A: Thanks for your suggestion. We have added description to the text as follow: “Although the association between F17 fimbriae and diarrhea is weaker than that is between F5 or F41 fimbriae and diarrhea, changes in epidemic trends demonstrate the necessity to consider F17 fimbriae in vaccine strategies.” Please see lines 50-53.
Line 67: You use the term 'F17G' here, but it is not stated that this is the adhesion subunit of F17 until the Discussion - please make this fact clear at this introductory stage.
A: The text is now corrected. Please see lines 69-70.
Line 113-4: You need to provide some detail on the conditions and apparatus used to lyse the bacteria under pressure. The phrase 'The fraction of inclusion body ...' is unclear. Please briefly describe how the cell debris was treated before affinity purification.
A: The text was nodified to “and the cells were broken at 800 bar for 5 cycles using a high-pressure homogenizer (Life Technologies, USA) after being suspended in PBS.” Please see lines 119-121.
Line 115: Suggest you describe the proteins as His-tagged here - it makes it easier for non-specialist readers to understand the affinity purification and the subsequent use of anti-his anibody for labelling.
A: Thanks for your suggestion. We have add to the description “His-tagged” presented here and results. Please see line 123, line 232 and line 238.
Line 117-8: "The purified proteins MEFA, rF17G, rFanC, rFim41a, and rSTa were purified ..." Is this just inappropriate repetition of 'purified' in this sentence, or was there an additional purification step?
A: Thank you for the comment. I’m sorry for this confusion. This is an inappropriate repetition, and the text is now corrected. Please see line 125.
Line 126: Why did you not use adjuvant without MEFA as your control injection, instead of PBS?
A: Thank you for the comment. In this study, we evaluated neutralizing antibodies against MEFA. Freunds adjuvant is a very routine choice in antibody evaluation, and its efficacy and safety are widely validated. Therefore, we did not set an additional Freund's adjuvant injection group here. In subsequent studies, multiple vaccine adjuvants will be considered for screening, and separate control injections will be performed for all adjuvants that are evaluated.
Line 127: "obtained and" is inappropriate here.
A: "obtained and" has been removed. Please see line 137.
Lines 133-4: I think 'coating' is more appropriate than 'coated' in both instances.
A: The text is now corrected. Please see lines 143-144.
Line 184: Was the STa you used the his-tagged recombinant protein you had purified, or did you obtain it elsewhere?
A: We used a recombinant STa protein without any tag. It was derived from earlier work in our laboratory.
Line 188: were the cells lysed with trypsin, as previously described?
A: Yes, the cells were lysed by trypsin. The text has been amended to “lysed with 0.25% trypsin as previously described”. Please see line 198.
Lines 223-5: This procedure is not described in Materials and Methods - you should transfer most of this description into M&M.
A: Thanks for your comments. We have transferred the specific description into the M&M. Please see lines 125-127.
Line 249: Fig 3 title should be expanded to include immunisation with FanC, Fim41a, F17G & STa, as well as MEFA.
A: Thanks again for your suggestion. The title of Fig. 3 has been changed to “Antibody detection of mice immunized with MEFA, rF17G, rFim41a, rFanC, and rSTa proteins.” Please see lines 257-258.
Line 250: suggest you retain only P<0.0001, as this is the only significance level marked.
A: The text is now corrected. Please see line 258, line 271 and line 284.
Line 258: Consider listing inhibition % to 2 sig figs (i.e. 93%, 84%, 88%) - it seems more appropriate and is more readable.
A: Thank you for your advice. We have listed inhibition % to Figure. 4. Please see line 267.
Line 263: as per comment for line 250
Figure 5: The diagonal hatching on the PBS control columns is difficult to distinguish from the hatching on the STa+preimmune serum columns. Suggest the PBS columns are a solid shade (grey, white or black).
A: The Figure. 5 has been altered. Please see line 284.
Line 277: Change "Antibodies neutralizated STa toxin " to 'Antibody neutralization of STa toxin".
A: Thanks again for your suggestion. The text has been changed to "Antibody neutralization of STa toxin". Please see line 285.
Lines 289-91: Again, you should acknowledge that the association between F17 and clinical disease is weaker than for F5 and F41.
A: Thanks again for your suggestion. The text has been modified as follow:
“Despite the association between F17 and clinical disease is weaker than for F5 and F41, this changing landscape underscores the necessity of reevaluating vaccine strategies, emphasizing newer and more prevalent fimbrial antigens to ensure the effectiveness of vaccines for preventing ETEC in calves.” Please see lines 299-302.
Line 321: Is this a technical (statistical) use of the word 'significantly' - if so, I can't see evidence of this in the results. If not, I advise using a different word, for clarity, e.g. 'substantially'.
A: The “significantly” in the text has been changed to “substantially”. Please see line 330.
Line 331: "The HCT-8 line is a suitable alternative" ... to what? To in vivo studies? To other cell lines? The sentence needs re-phrasing for clarity.
A: Thank you for the comment. I’m sorry for this confusion. The text has been modified to “the HCT-8 cell line is a more suitable alternative to other cell lines for in vitro studies.” Please see line 340.
Lines 357-361: This is accidentally-incorporated extraneous text, I think.
A: I’m sorry for this error. The accidentally-incorporated extraneous text has been removed.
Reviewer 2 Report
Comments and Suggestions for Authors
The authors constructed recombinant MEFA protein according to the four virulence factors, and its immunogenicity was detected by using several methods, however, there are some problems.
1. Line 53, “a favorable effect for calves” should be “a favorable effect on calves”.
2. The line spacing in lines 65-71 seems different from other paragraphs.
3. The authors constructed recombinant MEFA protein and immunized mice to collect serum, why not evaluate the protective effect of the immunization on ETEC-induced diarrhea in mice?
4. The subtitles of 3.4 and 3.5 should be modified.
5. There is no antibody neutralization activity described in section 3.5.
6. It seems that the authors forgot to delete the discussion template in lines 357-361.
Comments on the Quality of English Language
The quality of English language needs minor editing.
Author Response
Thank you for your comments. Below you will find our point-by-point responses:
The authors constructed recombinant MEFA protein according to the four virulence factors, and its immunogenicity was detected by using several methods, however, there are some problems.
- Line 53, “a favorable effect for calves” should be “a favorable effect on calves”.
A: Thank you for the comment. The text is now corrected. Please see line 55.
- The line spacing in lines 65-71 seems different from other paragraphs.
A: The text is now corrected. Please see lines 67-73.
- The authors constructed recombinant MEFA protein and immunized mice to collect serum, why not evaluate the protective effect of the immunization on ETEC-induced diarrhea in mice?
A: Thanks for your suggestion. This will be our next step. At present, the mouse model of E. coli is not mature enough for us to evaluate the E. coli causing diarrhea in calves by diarrhea symptoms in mouse models. The cattle model involves cost and animal welfare issues that we have to consider carefully. Our current work is focused on continuing to optimize the MEFA protein and to develop reliable alternative animal models.
- The subtitles of 3.4 and 3.5 should be modified.
A: Thanks again for your suggestion. The subtitle of 3.4 has been changed to "Adherence and Adherence Inhibition Assay", and the subtitle of 3.5 has been modified to "The activating impact and neutralizing properties of antibody". Please see line 263 and line 272.
- There is no antibody neutralization activity described in section 3.5.
A: Thank you for your comments. The increased level of cGMP is an important mechanism by which STa stimulates small intestinal cells. In the results, we quantified the stimulatory effect of STa toxin on cells by cGMP levels. However, after the STa toxin were neutralizied with anti-MEFA serum, cGMP levels were significantly decreased, which indirectly proved that the antibody had neutralizing effect on STa toxin. We made a little change to the text, please see line 274.
- It seems that the authors forgot to delete the discussion template in lines 357-361.
A: I’m sorry for this error, the discussion template has been deleted.
Reviewer 3 Report
Comments and Suggestions for Authors
This work explored a new vaccine candidate for ETEC.
It is a very good work but I do, however, have a few comments
1) the molecular biology beside the genetic constructs are not well described. It seems difficult to reproduce them with the information within material and methods. On one hand, there are PCR amplification of each ORF (where are the PCR conditions?) then cloning of individual ORF in PET30a plasmid (probably using NdeI=CATATG and SalI=GTCGAC enzymes), in the other hand there is the construction of MEFA by replacing minor epitopes in the F17G sequence by other epitopes. But finaly MEFA was not a genetic construction but a synthesis but there is no explanation about the cloning of the 1062 bp sequence into pET30? SO please, give more details about the construction of the recombinant plasmids.
2) As mentioned in the discussion, it would be good if your vaccine can be tested in cattle. Do you recommand to vaccinate the cows or the calves (or both)? Maybe a colostral protection could be enough because it concerns young calves.
3) Maybe, you could tested the vaccine in the neonatal mice model?
Carroll CJ, Hocking DM, Azzopardi KI, Praszkier J, Bennett-Wood V, Almeida K, Ingle DJ, Baines SL, Tauschek M, Robins-Browne RM. Re-evaluation of a Neonatal Mouse Model of Infection With Enterotoxigenic Escherichia coli. Front Microbiol. 2021 Mar 18;12:651488. doi: 10.3389/fmicb.2021.651488.Author Response
Thank you for your comments. Below you will find our point-by-point responses:
This work explored a new vaccine candidate for ETEC.
It is a very good work but I do, however, have a few comments
1) the molecular biology beside the genetic constructs are not well described. It seems difficult to reproduce them with the information within material and methods. On one hand, there are PCR amplification of each ORF (where are the PCR conditions?) then cloning of individual ORF in PET30a plasmid (probably using NdeI=CATATG and SalI=GTCGAC enzymes), in the other hand there is the construction of MEFA by replacing minor epitopes in the F17G sequence by other epitopes. But finaly MEFA was not a genetic construction but a synthesis but there is no explanation about the cloning of the 1062 bp sequence into pET30? SO please, give more details about the construction of the recombinant plasmids.
A: Thank you for your comments. We have realized that the description in Methods is not exhaustive enough, so we have included the following texts:
“The PCR conditions were as follows: An initial denaturation step at 98 °C for 2 min was followed by 30 cycles of the following conditions: 10 s at 98 °C, 5 s at 56 °C, and 10 s at 72°C. The amplification was completed with a final step of 72 °C for 1 min. The PCR products were verified by agarose gel electrophoresis and the band were cut out. Then these sequences were ligated separately to the pET30a vector digested by Nde I and Sal I enzymes using homologous recombination.” Please see lines 90-96.
In addition, MEFA is indeed a synthetic gene, and I am sorry for the trouble caused by our inappropriate expression. The text has been corrected to the following:
“Finally, a 1062 bp nucleotide sequence based on the model was synthesized into pET30a by Tsingke Biotec (Wuhan, China) The recombinant expression plasmid pET30a-F17G-Fim41a-FanC-STa was abbreviated as pET30a-MEFA.” Please see lines 109-111.
2) As mentioned in the discussion, it would be good if your vaccine can be tested in cattle. Do you recommand to vaccinate the cows or the calves (or both)? Maybe a colostral protection could be enough because it concerns young calves.
A: Thanks for your suggestion. This will be the further work. We believe that vaccinating heifers is a better protocol because diarrhea caused by ETEC occurs mostly in newborn calves without immunity. But experiments with cattle involve high costs and more animal welfare issues. We need to be careful about our next plans.
3) Maybe, you could tested the vaccine in the neonatal mice model?
A: Thanks again for your suggestion. In fact, we are also considering trying alternative animal models. At present, the mice model of E. coli is not mature enough for us to evaluate the E. coli from calves. Because E. coli could cause mice to die due to bacteremia by not diarrhea symptoms in mice models. The neonatal mice model may be a potential alternative.
Round 2
Reviewer 2 Report
Comments and Suggestions for Authors
No other comments.